# Polysaccharide-Enriched Fraction from *Amillariella Mellea* Fruiting Body Improves Insulin Resistance

**DOI:** 10.3390/molecules24010046

**Published:** 2018-12-22

**Authors:** Siwen Yang, Yuhan Meng, Jingmin Yan, Na Wang, Zhujun Xue, Hang Zhang, Yuying Fan

**Affiliations:** School of Life Sciences, Northeast Normal University, Jilin 130024, China; yangsw776@nenu.edu.cn (S.Y.); mengyh242@nenu.edu.cn (Y.M.); yanjm339@nenu.edu.cn (J.Y.); wangn324@nenu.edu.cn (N.W.); xuezj751@nenu.edu.cn (Z.X.); zhangh306@nenu.edu.cn (H.Z.)

**Keywords:** *Amillariella mellea*, polysaccharide, insulin resistance, lipid metabolism

## Abstract

Despite the edible fungus *Amillariella mellea* possessing a variety of biological activities, its effects on diabetes are still unclear. Polysaccharides are the main bioactive ingredients. In order to destroy the cell wall to obtain more polysaccharides, we used NaOH solution to extract *Amillariella mellea* fruiting bodies. The alkali extraction (AAMP) was identified as a polysaccharide-enriched fraction. Using type 2 diabetic rats induced by co-treatment of a high fat diet (HFD) and dexamethasone (DEX), we evaluated the hypoglycemic effects of AAMP. The results showed that oral administration of a high dose of AAMP markedly lowered fasting blood glucose, improving glucose intolerance and insulin resistance. AAMP also enhanced the level of LPL and the expressions of two critical lipases ATGL and HSL, leading to a decrease of serum triglyceride. In addition, AAMP specifically suppressed the expression of SREBP-1c, resulting in AAMP observably inhibiting lipid accumulation in the liver. These findings demonstrated that the improvement of AAMP on HFD/DEX-induced insulin resistance was correlated with its regulation of lipid metabolism. Our results indicated that AAMP could be a novel natural drug or health food used for the treatment of diabetes.

## 1. Introduction

Insulin resistance is defined as a reduced ability of tissues or cells to respond to normal levels of insulin, resulting in obesity and type 2 diabetes [1,2]. Accumulating studies suggest the relationship between lipid oversupply and insulin resistance [3,4]. Hyperinsulinemia may increase lipid synthesis and supply to muscles and the liver [5,6]. The Randle glucose-fatty acid cycle may contribute to the acute manifestation of insulin resistance in hyperlipidemic states, such as obesity [7]. Growth hormone, corticosteroids, and epinephrine are used to accelerate the release of fatty acids in adipose tissue, and through this cycle to inhibit muscle ingestion of glucose, leading to insulin insensitivity. In addition, excess glucocorticoid has been known to cause insulin resistance by inhibiting glucose transport [8].

Mushrooms are a potential source for functional foods due to their bioactive and non-toxic ingredients. Polysaccharides are mainly in the cell walls and the fruiting bodies of fungi [9]. *Amillariella mellea* is a popular edible fungus with wide distribution in the northeastern region of China. Previous studies have shown that the polysaccharides from *A. mellea* have potential biological activities, with a focus on anti-inflammation [10], anti-oxidation [11], and immunomodulation [12]. It has also been reported that polysaccharide from *A. mellea* promoted glucose-induced insulin secretion by scavenging the free radicals in alloxan-treated pancreatic cells [13]. However, it is not clear whether polysaccharide from *A. mellea* have a hypoglycemic effect in vivo.

We recently isolated a polysaccharide-enriched fraction (AAMP) from *Amillariella mellea* fruiting bodies. Here we evaluate the hypoglycemic activity of AAMP in a high-fat diet and glucocorticoid-induced diabetic rat and identify its mechanisms.

## 2. Results

### 2.1. Preparation of the Polysaccharide-Enriched Fraction from Amillariella Mellea Fruiting Body

The fruiting bodies of *Amillariella mellea* were defatted with 95% ethanol and extracted with hot water. The residues were further extracted with 0.5 M NaOH solution, precipitated with 75% ethanol, dialyzed with distilled water, and lyophilized; the extraction was named AAMP. It contained 68.4% total carbohydrates and 14.3% protein. The monosaccharide composition analysis showed AAMP was composed of glucose (Glc, 58.6%), galactose (Gal, 19.8%), mannose (Man, 18.1%), glucuronic acid (GlcA, 3.3%), and fucose (Fuc, 1.5%). The molecular weight distribution of AAMP was determined by HPGPC and two peaks were detected at different retention times. As shown in Figure 1, AAMP was composed of two fractions with molecular weights of 321 kDa and 23.3 kDa, respectively, and the latter was the major fraction.

### 2.2. Fourier Transform Infrared Spectroscopy (FT-IR) Analysis

The FT-IR spectrum of AAMP exhibited typical carbohydrate patterns (Figure 2). The intense and broad peak around 3389 cm^−1^ indicated the presence of O-H stretching in hydrogen bonds [14]. The weak band near 2922 cm^−1^ was attributed to the C-H stretching and bending vibrations. The carbonyl groups (C=O) were also observed with characteristic absorptions at the asymmetrical stretching band at 1636 cm^−1^ and symmetric stretching band around 1400 cm^−1^ [15]. The peak at 1076 cm^−1^ was assigned to the O-H variable angle vibration, and the characteristic band near 1043 cm^−1^ suggested the presence of pyranose rings. Moreover, the weak peak at 915 cm^−1^ indicated the presence of β-linked-d-glucopyranose residues [16].

### 2.3. AAMP Reduced Fasting Blood Glucose in HFD/DEX-Treated Rat

The hypoglycemic effect of AAMP was examined by high-fat diet and dexamethasone co-treated (HFD/DEX) rats. Compared with the body weight of normal rats, HFD/DEX treatment slowed the body weight increase. AAMP did not alter the body weight change of HFD/DEX-treated rats (Figure 3A). As shown in Figure 3B, the fasting blood glucose (FBG) of rats treated with HFD/DEX exerted higher than that of normal rats, suggesting that HFD/DEX might cause insulin resistance. In normal rats, a high dose of AAMP did not affect FBG, but in HFD/DEX-treated rats, AAMP slightly reduced FBG at a low dose (50 mg/kg/day), and significantly reduced FBG at a high dose (200 mg/kg/day). The results indicated that AAMP specifically reduced blood glucose in insulin-resistant rats.

### 2.4. AAMP Improved Glucose Intolerance in HFD/DEX-Treated Rat

During glucose tolerance test (GTT), the blood glucose levels of rat treated with HFD/DEX were higher at all considered times compared to normal rats (Figure 4), indicating that HFD/DEX generated glucose intolerance. In HFD/DEX-treated rats, AAMP caused the blood glucose to rise and then fall back quickly to the initial level at a high dose (Figure 4A), and reduced the AUC (Figure 4B), indicating that it obviously improves glucose intolerance.

### 2.5. AAMP Ameliorated Insulin Resistance in HFD/DEX-Treated Rat

The serum insulin of rat after HFD/DEX treatment was distinctly higher than that of normal rat (Figure 5A), and the HOMA-IR (homeostasis model assessment index for insulin resistance) of the former was also higher than the latter (Figure 5B), suggesting that HFD/DEX caused severe insulin resistance. After administration of AAMP, the serum insulin decreased (Figure 5A) and the HOMA-IR declined (Figure 5B) compared to HFD/DEX-treated rats. These results demonstrated that AAMP improved insulin resistance.

### 2.6. AAMP Lowered Serum Triglyceride and Free Fatty Acids in HFD/DEX-Treated Rat

Abnormal lipid metabolism usually causes insulin resistance, so we examined whether AAMP affected lipid metabolism. The serum triglyceride (TG),total cholesterol (TC), and free fatty acids (FFA) of rats after HFD/DEX treatment were increased than those of normal rats (Figure 6A–C), indicating that HFD/DEX affected lipid metabolism. After AAMP administration, the serum TG and FFA significantly decreased, but not TC, compared to HFD/DEX-treated rats (Figure 6A–C).

Lipoprotein lipase (LPL) is important in lipid metabolism, so we measured LPL contents. The effect of HFD/DEX on LPL was not significant compared to the control diet, however, AAMP could enhance the level of LPL (Figure 6D), indicating that the significant increase in the LPL level could partly be a reason for the decreased serum TG level, and that AAMP probably regulated lipid metabolism in other manners.

### 2.7. AAMP Enhanced Lipolysis and Suppressed Lipogenesis

Since AAMP reduced the contents of serum TG and FFA, we explored the mechanisms of AAMP to regulate lipid metabolism. Adipose triglyceride lipase (ATGL) and hormone-sensitive lipase (HSL) are the two important lipases. AAMP increased the expressions of these two lipases compared to HFD/DEX-treated rats (Figure 7A), indicating that AAMP enhanced lipolysis to decrease TG content. Compared to the normal rat, the muscle cells of the rat became smaller and the cell boundary was blurred after HFD/DEX treatment, and AAMP did not improve rat muscle damage caused by HFD/DEX (Figure 7B). However, AAMP significantly inhibited lipid droplet accumulation in liver (Figure 7B). In addition, the effects of AAMP on lipogenesis transcription factor expressions in liver were also examined. AAMP significantly reduced the expression of SREBP-1c, but no effect on the expression of PPARγ (Figure 7C), suggesting that AAMP inhibited lipogenesis by regulation of transcription factor SREBP-1c.

## 3. Discussion

Long-term feeding with a HFD may increase fat accumulation and high blood glucose in rats [17]. Dexamethasone is a glucocorticoid frequently used to induce insulin resistance by enhancing hepatic glucose output and suppressing the peripheral glucose uptake [18]. Herein, the rats co-treated with HFD and DEX exhibited a loss of body weight, and an increase of fasting blood glucose, serum insulin levels, and triglycerides, which are typical features of type 2 diabetes [19]. The indicators of HFD/DEX-treated rats observed in our experiments are consistent with the literature, indicating that we successfully established a rat model of type 2 diabetes. The polysaccharide-enriched fraction AAMP from *Amillariella mellea* fruiting body exerts an oral hypoglycemic effect at high dose by lowering fasting blood glucose, improving glucose-intolerance, and reducing serum insulin, TG, and FFA levels.

Dexamethasone treatment is associated with an increase of triglyceride production rates and hypertriglyceridemia [20]. The triglycerides are hydrolyzed step by step to produce free fatty acid and glycerol. We found that AAMP increases the level of LPL and the expressions of two main lipases, ATGL and HSL, in HFD/DEX-treated rat, resulting in a decrease of triglycerides. Fatty acids through the bloodstream are absorbed by liver or muscle to participate in lipid metabolism. AAMP specifically reduces the number of lipid droplets in liver, but has no effect on fat deposition in muscle. It is speculated that AAMP also affects lipogenesis in the liver. Lipogenesis is controlled by multiple transcriptional regulators, such as SREBP and PPAR. SREBP-1c activates genes involved in fatty acid and TG synthesis. ER stress stimulates the activation of SREBP-1c to increase in lipogenesis [21]. PPARγ controls fatty acid uptake, trafficking, and TG synthesis, ablation of hepatic PPARγ alleviates liver steatosis in obese mice [22,23]. AAMP specifically suppressed the expression of SREBP-1c, but not PPARγ, indicating that AAMP probably regulated fatty acid and TG synthesis. Therefore, the hypoglycemic effect of AAMP is likely related to increased lipolysis and decreased lipogenesis.

AAMP could regulate lipid metabolism to lower blood glucose in hyperglycemic rats and had no toxic side effects on normal rats, suggesting that AAMP has the potential to develop hypoglycemic health foods and drugs.

## 4. Materials and Methods

### 4.1. Preparation of the Polysaccharide-Enriched Fraction

Fruiting bodies of *Amillariella mellea* were collected from Changbai Mountain in Jilin Province, China and identified using rDNA-ITS sequencing analysis. Fruiting bodies were defatted with 95% ethanol, and the residues were extracted with distilled water three times at 100 °C for 4 h. After filtration, the residues were dried and extracted using 0.5 M NaOH/NaBH_4_ at 80 °C. Extracts were neutralized with glacial acetic acid, and precipitated with 75% ethanol. The precipitate was collected and lyophilized. The alkali extraction obtained from *Amillariella mellea* was named AAMP. The total carbohydrate content, protein content, and monosaccharide composition were also detected according to classical methods [24,25,26]. The molecular weight distribution of AAMP was determined by using gel-permeation chromatography on a Shimadzu HPLC system (Shimadzu, Tokyo, Japan) coupled with a TSK-gel G-3000PWXL column (7.8 × 300 mm, TOSOH, Tokyo, Japan) as described by Zhang [27]. The column was pre-calibrated by using standard dextrans (50 kDa, 25 kDa, 12 kDa, 5 kDa, and 1 kDa) with linear regression [27].

### 4.2. Fourier Transform Infrared Spectroscopy Analysis

Structures of polysaccharides, such as glycosidic bonds and functional groups, can be analyzed using the FT-IR spectroscopy. FT-IR spectrum of AAMP was determined on a Spectrum Two spectrometer (PerkinElmer, Waltham, MA, USA) in a range of 4000–400 cm^−1^. The sample was ground with KBr powder and then pressed into 1 mm pellets for FT-IR measurements.

### 4.3. Experimental Protocols

Male Sprague Dawley rats (SYXK 2018-0015) (180–200 g) were obtained from Model Animal Research Center of Nanjing University (Nanjing, China). The animals were housed in a room maintained at 23 ± 2 °C with relative air humidity of 45% to 55% on a 12 h light/12 h dark cycle. Before treatment, the fasting blood glucose (FBG) and oral glucose tolerance of the rats were examined as the basic values. The rats were divided into six groups (*n* = 15), including the control group, control plus high dose of AAMP (200 mg/kg/day) group, model group, model plus low dose of AAMP (50 mg/kg/day) group, model plus moderate dose of AAMP (100 mg/kg/day) group, and model plus high dose of AAMP (200 mg/kg/day) group. The control groups were given normal chow diet (10% kcal% fat). The model groups were continuously fed with 60% kcal% fat diet (HFD, Research Diets, D12492) for 35 days, and dexamethasone (0.8 mg/kg, i.p) was additionally administered on the last 10 days. The rats treated with AAMP were intragastrically administered for 35 days, once daily.

### 4.4. Fasting Blood Glucose Test

The rats were fasted for 4 h. The blood was collected from tail and measured by the One Touch Ultra Easy Glucometer (Johnson, New Brunswick, NJ, USA).

### 4.5. Glucose Tolerance Test (GTT)

The rats were fasted for 4 h followed by intragastrically administered with 2.5 g/kg body weight of glucose. Blood glucose levels were monitored at 0, 30, and 120 min after administration using blood collected from the tail using the One Touch Ultra Easy Glucometer.

### 4.6. Biochemical Indicators Detection

The serum level of insulin was measured using a commercially available kit (IBL international, Hamburg, Germany). The homeostasis model assessment index for insulin resistance (HOMA-IR) was defined as (FBG × serum insulin)/22.5. The contents of serum triglyceride (TG), total cholesterol (TC), and free fatty acid (FFA) were measured by the kits (Jiancheng, Nanjing, China). The content of serum lipoprotein lipase (LPL) was measured using an ELISA kit (AMEKO, Shanghai, China).

### 4.7. H and E Staining

The liver and muscle were fixed in 4% paraformaldehyde (Sangon Biotech, A500684-0500, Shanghai, China), and then embedded in paraffin, and sectioned at 5 μm. H and E staining was performed following standard protocols. Photos were taken using a light microscope (OLYMPUS BX51, Tokyo, Japan).

### 4.8. Western Blots

The adipose tissues were lysed with lysis buffer containing 50 mM Tris (pH 7.4), 150 mM NaCl, 1 mM EDTA, 1% Triton X-100, proteinase inhibitor cocktail (Roche Applied Sciences, Penzberg, Upper Bavaria, Germany) and phosphatase inhibitor cocktail (Thermo Scientific, Waltham, MA, USA). Western blots were performed as described in the previous publication [24] using the following primary antibodies: anti-ATGL (Abcam, Cambridge, UK, ab109251, 1:1000), anti-HSL (Sigma, Milpitas, CA, USA, SAB4501763, 1:1000), and anti-actin (Abcam, 1:1000) antibodies.

### 4.9. Quantitative Real-Time PCR (RT-qPCR)

Total RNA of the tissues was extracted using Trizol (Thermo Fisher, Waltham, MA, USA) according to the instructions provided by the manufacturer. cDNA was generated from 1 μg of RNA using M-MLV reverse transcriptase (Promega, Fitchburg, WI, USA). qPCR was performed using SYBR Green real-time PCR master mixes (Thermo Fisher, Waltham, MA, USA) and a LightCycler 480 real-time PCR system (Roche Applied Science, Indianapolis, IN, USA). qPCR primer sequences are listed in Table 1.

### 4.10. Statistical Analysis

The results were expressed as the mean ± S.E.M. Statistical analysis of the data was performed using Student’s t-test and two-way repeated-measures ANOVA with Dunnett’s post-hoc test (IBM SPSS Statistics 17.0, Armonk, NY, USA). Differences were considered significant when *p* < 0.05.

## Figures and Tables

**Figure 1 molecules-24-00046-f001:**
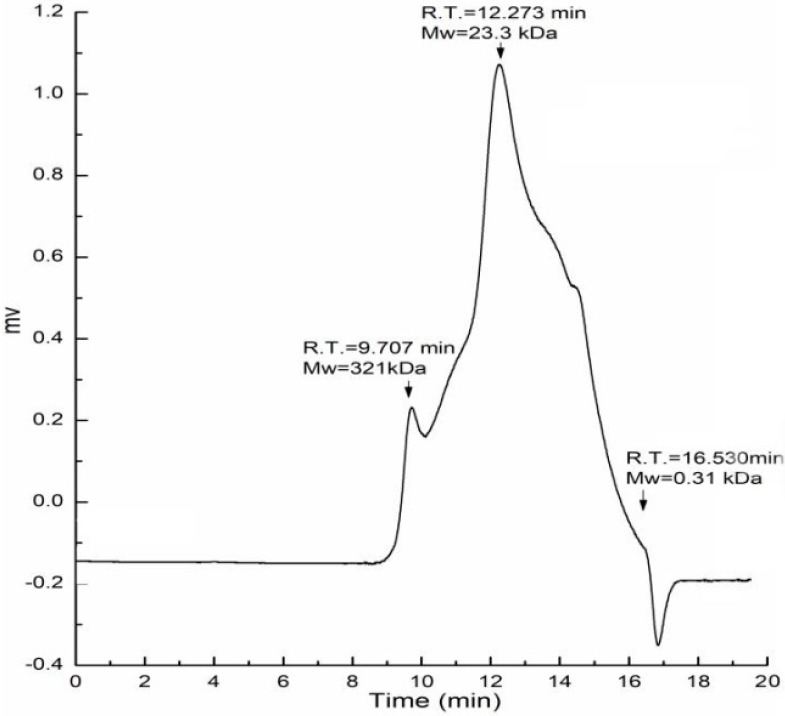
Molecular weight distribution of AAMP.

**Figure 2 molecules-24-00046-f002:**
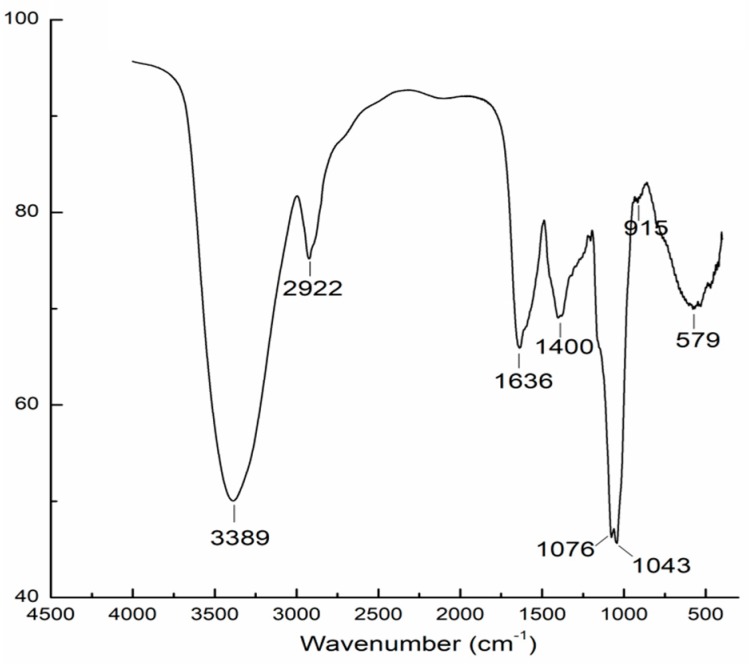
FT-IR spectrum of AAMP.

**Figure 3 molecules-24-00046-f003:**
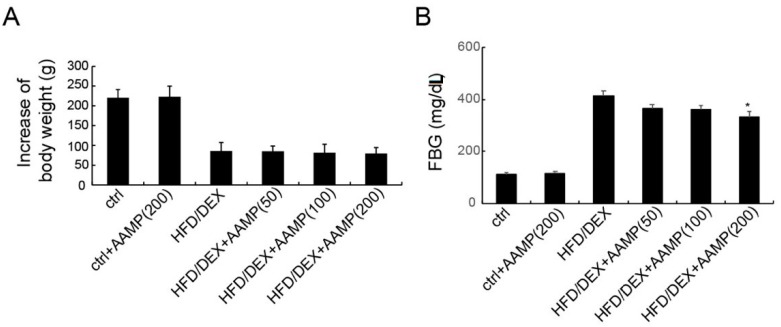
AAMP treatment lowered blood glucose of the diabetic rats. The rats were administered with different doses of AAMP for 35 days. (**A**) Increase of body weight. (**B**) Fasting blood glucose. Results represent the mean ± S.E.M. (*n* = 15 rats in each group). * *p* < 0.05 compared to HFD/DEX rats.

**Figure 4 molecules-24-00046-f004:**
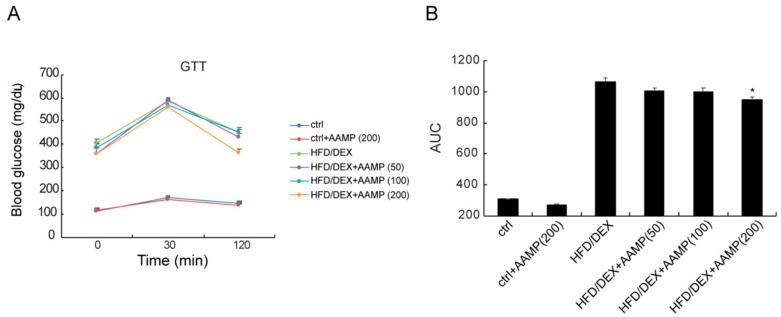
AAMP treatment ameliorated glucose intolerance of the diabetic rats. At the last day of the animal experiment, blood glucose changes were observed at 0, 30, and 120 min after intragastric administration of glucose in rats. (**A**) GTT. (**B**) The calculation of area under the curve (AUC) from GTT. Results represent the mean ± S.E.M. (*n* = 15 rats in each group). * *p* < 0.05 compared to HFD/DEX rats.

**Figure 5 molecules-24-00046-f005:**
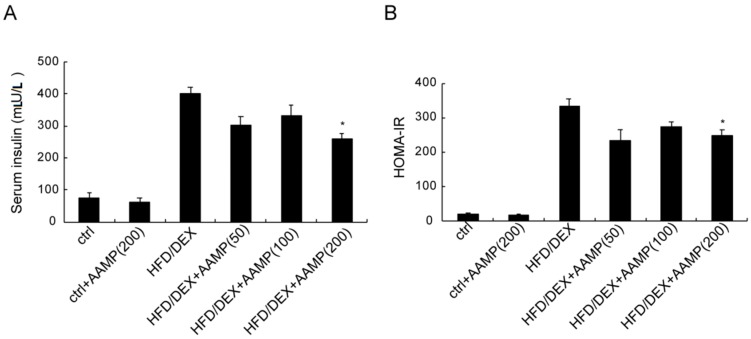
AAMP treatment enhanced insulin sensitivity of the diabetic rats. (**A**) The content of serum insulin. (**B**) The calculation of HOMA-IR from FBG and serum insulin. Results represent the mean ± S.E.M. (*n* = 15 rats in each group). * *p* < 0.05 compared to HFD/DEX rats.

**Figure 6 molecules-24-00046-f006:**
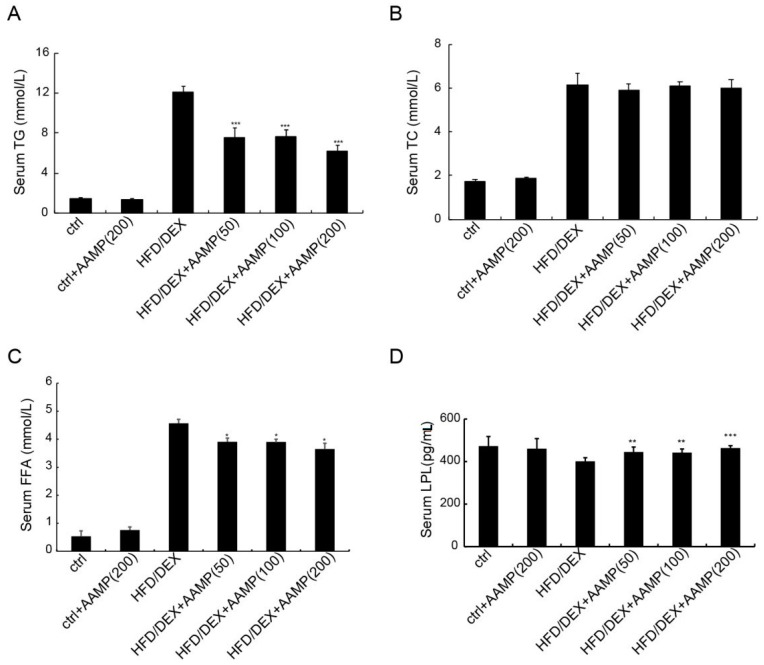
Metabolic profiles of diabetic rats treated with AAMP. (**A**) Triglyceride (TG). (**B**) Total cholesterol (TC). (**C**) Free fatty acid (FFA). (**D**) Lipoprotein lipase (LPL). Results represent the mean ± S.E.M. (*n* = 15 rats in each group). * 0.01 < *p* < 0.05; ** 0.001 < *p* < 0.01; *** *p* < 0.001 compared to HFD/DEX rats.

**Figure 7 molecules-24-00046-f007:**
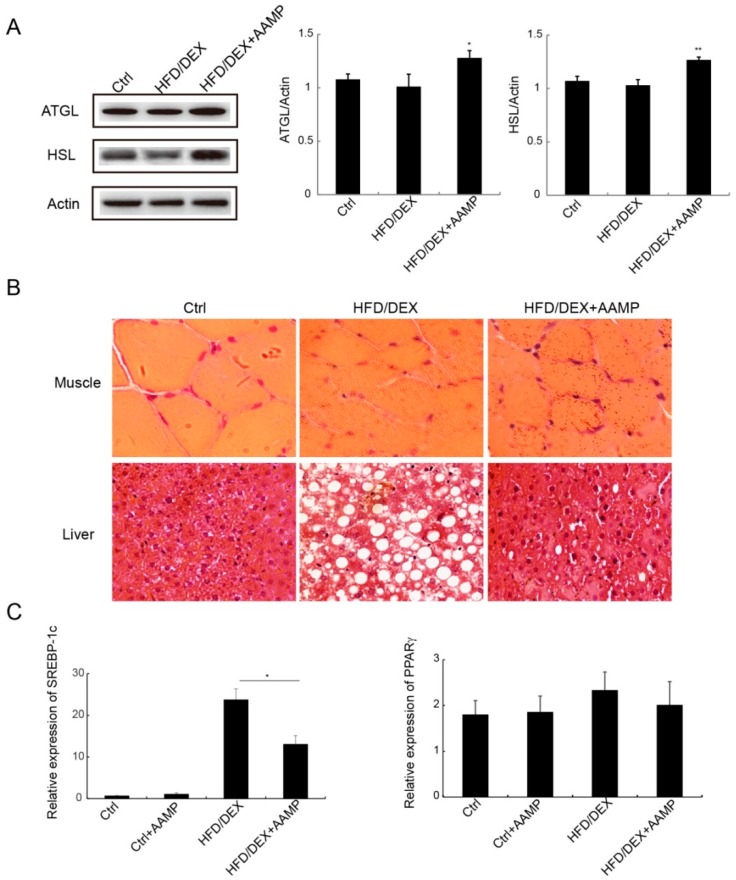
Effects of AAMP on lipid metabolism in diabetic rats. (**A**) Expressions of lipases in adipose tissue (**left**) and quantitative analysis (**right**). (**B**) The lipid droplets in liver and muscle of the rats. (**C**) The lipogenesis-related gene expressions in liver. SREBP-1c, sterol regulatory element binding protein 1c; PPARγ peroxisome proliferator-activated receptor γ Results represent the mean ± S.E.M. (*n* = 15 rats in each group). * 0.01 < *p* < 0.05; ** 0.001 < *p* < 0.01 compared to HFD/DEX rats.

**Table 1 molecules-24-00046-t001:** The primers used in our study.

Gene Symbol	Forward (5′-3′)	Reverse (5′-3′)
SREBP-1c	GGAGCCATGGATTGCACATT	GCTTCCAGAGAGGAGGCCAG
PPARγ	CCCTTTACCACGGTTGATTTCTC	GCAGGCTCTACTTTGATCGCACT
GAPDH	ATGATTCTACCCACGGCAAG	CTGGAAGATGGTGATGGGTT

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
