# Peer review of "Polysaccharide-Enriched Fraction from Amillariella Mellea Fruiting Body Improves Insulin Resistance"

_molecules, 2018, doi:10.3390/molecules24010046_

Round 1

Reviewer 1 Report

 This manuscript described a role of polysaccharide-enriched fraction from Amillariella mellea fruiting body on insulin resistance. An overall effect of AAMP on glucose metabolism is quite mild although AAMP may affect lipid metabolism more significantly. This work would eventually have a scientific merit; however, there are issues which should be addressed properly by authors in the current form.

Major Issues:

1.     No information on diet composition of HFD as well as control diet was provided.

2.     Rats fed HFD/DEX gain much less weight than those on control diet and showed severe hyperglycemia compared to controls in Figure 3.

Please show how the reduced body weight gain in rats on HFD/DEX compared with those on control diet began and how the onset associated with their diabetic states, food intake, and/or energy expenditure during the course of the experiment. Rat on HFD alone for 25 days gain more weight than those on control diet? To evaluate the effect of DEX alone in control diet and how AAMP modifies the outcome, did the authors do the experiments with rats fed control diet/DEX? How did they choose the dose of DEX at the concentration of 0.8mg/kg i.p. for the last 10 days of the experiment? How did AAMP treatment affect food intake and energy expenditure in both rats on control diet and HFD/DEX?

3.     Serum TG levels are mainly regulated by lipoprotein lipase (LPL). Did the authors measure LPL activity? How did the serum lipoprotein component change in the rats on HFD/DEX vs. control diet? Did the authors test serum TG clearance in those rats?

4.     Did the authors measure FFA levels?

5.     Quantitative analysis is required for Figure 7A to make the authors’ point.

6.     In line 147 it is not clear in what triglyceride decreased.

Minor Issues:

7.     Minor English editing is necessary throughout the manuscript.

8.     FT-IR should be spelled out in line 58.

Author Response

Thanks for considering our manuscript and all of the useful comments. The reviewer’s comments are very important for our future research, so we revised the manuscript, and gave responses to the comments as follows.

Reviewer: This manuscript described a role of polysaccharide-enriched fraction from Amillariella mellea fruiting body on insulin resistance. An overall effect of AAMP on glucose metabolism is quite mild although AAMP may affect lipid metabolism more significantly. This work would eventually have a scientific merit; however, there are issues which should be addressed properly by authors in the current form.

Major Issues:

1. No information on diet composition of HFD as well as control diet was provided.

Response to comments: We have provided the informations of HFD as well as control diet in lines 196-197.

2. Rats fed HFD/DEX gain much less weight than those on control diet and showed severe hyperglycemia compared to controls in Figure 3. Please show how the reduced body weight gain in rats on HFD/DEX compared with those on control diet began and how the onset associated with their diabetic states, food intake, and/or energy expenditure during the course of the experiment. Rat on HFD alone for 25 days gain more weight than those on control diet? To evaluate the effect of DEX alone in control diet and how AAMP modifies the outcome, did the authors do the experiments with rats fed control diet/DEX? How did they choose the dose of DEX at the concentration of 0.8mg/kg i.p. for the last 10 days of the experiment? How did AAMP treatment affect food intake and energy expenditure in both rats on control diet and HFD/DEX?

Response to comments:

The purpose of this project is to screen an active fraction from Amillariella mellea fruiting body that lowers blood glucose and develop it into an oral hypoglycemic health food. Therefore, we established a diabetic rat model according to “National Handbook for Implementing Technical Specification for Evaluation of Health Food”, and choose the 0.8 mg/kg i.p. of DEX for the last 10 days. Based on the references [1-3] and handbook, the rats showed hyperinsulinemia after HFD/DEX treatment, and our result of serum insulin content was consistent with the references (Figure 5). During the course of the experiment, the food intake was no significant changes between control rats and diabetic rats, suggesting AAMP did not affect food intake in rats. But energy expenditure was still unknown at present due to lack of the examination tools. The increase of body weight gain in rats fed with HFD alone was similar to that in rats fed with control diet.

In fact, a diabetic rat model can also be established if HFD is fed alone for a long period of time (at least 12 weeks). However, short-term treatment of HFD and DEX can accelerate insulin resistance in rats and establish diabetic rat model faster. The reviewer's suggestions are very meaningful, and we perhaps examine the effect of AAMP on DEX alone treated rats in subsequent trials.

References:

[1] Severino, C.; Brizzi, P.; Solinas, A.; Secchi, G.; Maioli, M.; Tonolo, G. Low-dose dexamethasone in the rat: a model to study insulin resistance. Am J Physiol Endocrinol Metab 2002, 283, E367-E373.

[2] Caro, J.F.; Amatruda, J. M. Glucocorticoid-induced insulin resistance. J Clin Invest 1982, 69, 866-875.

[3] Hettiarachchi, M.; Watkinson, A.; Jenkins, A.B.; Theos, V.; Ho, K.Y. Kraegen, E.W. Growth hormone-induced insulin resistance and its relationship to lipid availability in the rat. Diabetes 1996, 45, 416-421.

3. Serum TG levels are mainly regulated by lipoprotein lipase (LPL). Did the authors measure LPL activity? How did the serum lipoprotein component change in the rats on HFD/DEX vs. control diet? Did the authors test serum TG clearance in those rats?

Response to comments:

Based on the reviewer’s suggestions, we have measured LPL activity (Figure 6D). The serum LPL of diabetic rats was lower than that of control rats, but AAMP treatment could increase the LPL level.

We examined the contents of LDL and HDL in rats of control and HFD/DEX. As shown in the table below, HFD/DEX treatment significantly increased the levels of LDL and HDL. HFD/DEX increased the contents of TG, TC and FFA, resulting in increased lipoprotein production. Our results showed that AAMP could decrease serum TG and FFA (Figure 6A and C), but the clearance of TG was still not detected so far.

Control (mmol/L)

HFD/DEX (mmol/L)

LDL

8.40±1.22

38.2±0.83 ***

HDL

6.06±2.12

45.98±1.57 ***

4.     Did the authors measure FFA levels?

Response to comments: We have measured FFA level (Figure 6C). HFD/DEX treatment increased FFA level compared to the control diet, but AAMP treatment could significantly decrease the FFA level.

5.     Quantitative analysis is required for Figure 7A to make the authors’ point.

Response to comments: We have added the quantitative analysis in Figure 7A.

6.     In line 147 it is not clear in what triglyceride decreased.

Response to comments: We have revised the description “reducing serum insulin, TG and FFA levels” in line 150.

Minor Issues:

7.     FT-IR should be spelled out in line 58.

Response to comments: We have revised the description in line 59.

Reviewer 2 Report

Review attached.

Author Response

Thanks for considering our manuscript and all of the useful comments. The reviewer’s comments are very important for our future research, so we revised the manuscript, and gave responses to the comments as follows.

Reviewer 2: The present study prepared an alkali polysaccharide-enriched fraction (AAMP) from the edible fungus A. mellea. The hyperglycemic effect of AAMP was examined by high fat diet and dexamethasone co-treated (HFD/DEX) rat. The results showed that oral administration of high dose of AAMP markedly lowered fasting blood glucose, improving glucose intolerance and insulin resistance. AAMP also enhanced the expressions of two critical lipases ATGL and HSL, leading to a decrease of serum triglyceride. In addition, AAMP suppressed the expression of SREBP-1c, resulting in AAMP observably inhibited lipid accumulation in liver. However, the present research work only displayed preliminary advance and will not have significant impacts in the molecule-based medicines related research field. Therefore, this manuscript is not recommended to accept for publication in Molecules. In addition, there are some major comments to be addressed as following.

1. There were some minor typographic, grammar, and format errors to be found in the text, such as lines 38, 40, 42, 70, etc. Authors have to check and revise these errors.

Response to comments: We have checked typographic, grammar, and format in line 38, 40, 42 and 70, and through the manuscript.

2. In the bioactivity section, authors have to provide the data of positive control. According to Figures 3-6, the bioactivity data were not significant. The examined concentration is too high to make any readers interested. Lines 75-76, 84-88, the experimental results were not so significant and sentences provided by authors were overclaimed.

Response to comments:

We agree with the reviewer’s point that we should provide the positive control. In the follow-up study, we continued to fractionate AAMP to obtain a polysaccharide with uniform molecular weight. Currently, the polysaccharide is undergoing oral hypoglycemic effect detection and we used metformin as a positive control drug. However, due to time we can't make up metformin in this manuscript.

Regarding the high dose, we think that 200 mg/kg is a medium dose. The commercially available oral hypoglycemic drug metformin has a daily dose of 0.5-1 g (assume human body weight is 60 kg). The equivalent dose of the rat is 6.3 times that of the human body, which means that the oral metformin should be administered to the rats at 105 mg/kg. The molecular weight of metformin is approximately 165 Da. The AAMP dose is 200 mg/kg but the molecular weight is approximately 23.3 kDa. Therefore, the molar concentration of AAMP is very low compared to metformin.

3. In the References section, the writing manner of some references did not follow the style of this journal. Authors have to check and revise these errors.

Response to comments: We have checked and revised the references.

Reviewer 3 Report

This study analyzes the effects of a mushroom extract on an animal model insulin resistance in rat. The results indicate that the extract improve glucose tolerance probably by changes in lipid metabolism.

The student t test is not appropriate when several groups are compared, as in figures 3 to 6. A multiple comparison test, such as Dunnett or Bonferroni, should be used for statistical analysis. Some of the changes observed are small, and it is possible that they will not be significant when a more exigent test is used.

Quantitative data of the western experiments should be provided, not just a representative blot.

The composition, or at least the reference, of the high fat diet should be included

Author Response

Thanks for considering our manuscript and all of the useful comments. The reviewer’s comments are very important for our future research, so we revised the manuscript, and gave responses to the comments as follows.

Reviewer: This study analyzes the effects of a mushroom extract on an animal model insulin resistance in rat. The results indicate that the extract improve glucose tolerance probably by changes in lipid metabolism.

1. The student t test is not appropriate when several groups are compared, as in figures 3 to 6. A multiple comparison test, such as Dunnett or Bonferroni, should be used for statistical analysis. Some of the changes observed are small, and it is possible that they will not be significant when a more exigent test is used.

Response to comments: The reviewer's suggestions are very helpful to our study, and we should pay attention to combine the single factor test and the multiple comparison test when doing data analysis. In the next study we will increase the number of samples and consider multiple comparison test.

2. Quantitative data of the western experiments should be provided, not just a representative blot.

Response to comments: We have added the quantitative analysis in Figure 7A.

3. The composition, or at least the reference, of the high fat diet should be included.

Response to comments: We have provided the information of HFD in lines 193-194.

Round 2

Reviewer 1 Report

In lines 113-116, the authors did not measure serum LPL activity but concentration (mass). Please correct them.

In the quantitative data of ATGL and HSL, there is no change in those protein levels between rats in control and HFD/DEX as well as LPL levels in serum. However, the authors only highlighted the significant increase of ATGL and HSL levels in HFD/DEX +AAMP to explain the decrease in lipid droplets in the liver. Significant increase LPL mass could partly be a reason the decreased serum TG levels in AAMP-treated animals?

Author Response

Response to comments: We have corrected the description in lines 117-120. Besides that, we absolutely agree with the reviewers' opinion. AAMP does regulate lipid metabolism in multiple manners, so we re-described our points of view in lines 18-19, 119-120 and 160-161.

Reviewer 2 Report

I had gone through the revised manuscript carefully and this manuscript was recommended for acceptance after minor revisions since authors had provided suitable explanation for part of the previous queries. The following points should be revised and checked.

In the bioactivity section, there was still no positive control in this assay.

I believed that authors have to elucidate the main composition of AAMP more detail rather than only analysis of the monosaccharide composition.

Author Response

Reviewer: In the bioactivity section, there was still no positive control in this assay.

Response to comments:

We absolutely agree with reviewer’s opinion. The purpose of our project is to screen an active fraction from Amillariella mellea fruiting body that lowers blood glucose and develop it into an oral hypoglycemic health food. Therefore, we designed our study according to “National Handbook for Implementing Technical Specification for Evaluation of Health Food”. In the handbook, there is no requirement for a positive control drug, so we did not use a positive control in this article.

However, we also realize the importance of positive control. Based on the reference (Biochemical Pharmacology, 56, 1145-1150, 1998), Metformin could reduce the extent of dexamethasone-induced hyperglycaemia and decreased insulin resistance. Thus, metformin is a suitable positive control drug in our study. In the experiment we are currently working on, a positive control with metformin has been added, but no results have been obtained yet.

Reviewer: I believed that authors have to elucidate the main composition of AAMP more detail rather than only analysis of the monosaccharide composition.

Response to comments:

AAMP contained 68.4% of total carbohydrate, 14.3% of protein and other impurities including ash. From the result of molecular weight, AAMP was composed of two fractions. Since AAMP is a mixed polysaccharide, it makes little sense to directly obtain more structural information.

In a subsequent experiment, we further fractionated AAMP using a DEAE-cellulose column to obtain two homogeneous polysaccharides AAMP-N and AAMP-A. After structural analysis, AAMP-N is a mannogalactoglucan, AAMP-A is glucan. And the hypoglycemic effects of them were still under investigation, thus these data are not suitable for inclusion in this article.

Reviewer 3 Report

There are still concerns over the statistical analysis. A post hoc multiple comparison test such as Dunnett or Bonferroni is necessary when several groups are compared, as in this work. Without that, the significance of the results cannot established with confidence, and their interpretation may be doubtful  

Author Response

Reviewer: There are still concerns over the statistical analysis. A post hoc multiple comparison test such as Dunnett or Bonferroni is necessary when several groups are compared, as in this work. Without that, the significance of the results cannot be established with confidence, and their interpretation may be doubtful. 

Response to comments: According to reviewer’s suggestion, we have re-done statistical analysis in Figure 3-7 using two-way repeated-measures ANOVA with Dunnett’s post-hoc test (SPSS Statistics 17.0). We confirmed that the significance of the results is credible. 
